# Annealing-Driven Microstructural Evolution and Its Effects on the Surface and Nanomechanical Properties of Cu-Doped NiO Thin Films

**San-Ho Wang [1], Sheng-Rui Jian [1],\*, Guo-Ju Chen [1],\*, Huy-Zu Cheng [1],\* and Jenh-Yih Juang [2]**

[1] Department of Materials Science and Engineering, I-Shou University, Kaohsiung 840, Taiwan; sanho.wang@yahoo.com.tw

[2] Department of Electrophysics, National Chiao Tung University, Hsinchu 300, Taiwan; jyjuang@g2.nctu.edu.tw

\* Correspondence: srjian@gmail.com (S.-R.J.); gjchen@isu.edu.tw (G.-J.C.); huyzu@isu.edu.tw (H.-Z.C.); Tel.: +886-7-6577711-3130 (S.-R.J.); +886-7-6577711-3116 (G.-J.C.); +886-7-6577711-3117 (H.-Z.C.)

**Abstract:** The effects of annealing temperature on the structural, surface morphological and nanomechanical properties of Cu-doped (Cu-10 at %) NiO thin films grown on glass substrates by radio-frequency magnetron sputtering are investigated in this study. The X-ray diffraction (XRD) results indicated that the as-deposited Cu-doped NiO (CNO) thin films predominantly consisted of highly defective (200)-oriented grains, as revealed by the broadened diffraction peaks. Progressively increasing the annealing temperature from 300 to 500 °C appeared to drive the films into a more equiaxed polycrystalline structure with enhanced film crystallinity, as manifested by the increased intensities and narrower peak widths of (111), (200) and even (220) diffraction peaks. The changes in the film microstructure appeared to result in significant effects on the surface energy, in particular the wettability of the films as revealed by the X-ray photoelectron spectroscopy and the contact angle of the water droplets on the film surface. The nanoindentation tests further revealed that both the hardness and Young's modulus of the CNO thin films increased with the annealing temperature, suggesting that the strain state and/or grain boundaries may have played a prominent role in determining the film's nanomechanical characterizations.

**Keywords:** Cu-doped NiO thin films; XRD; surface energy; nanoindentation; hardness

## 1. Introduction

NiO is a p-type semitransparent conducting oxide with reported bandgap ranging from 3.6 to 4.0 eV [1]. Combining with its excellent chemical stability and optoelectronic properties, NiO films have been considered as a promising candidate for various applications, such as solar cell [2,3], electrochromic display devices [4], sensors [5,6], and a wide variety of electronic devices [7,8]. Although stoichiometric NiO is an insulator with a high resistivity of ~$10^{13}$ Ω-cm at room temperature, its resistivity can be significantly reduced via creating substantial concentrations of nickel vacancies and forming interstitial oxygen atoms in NiO crystallites [9]. Recently, it was demonstrated that by doping metallic elements, such as Li [10,11], Na [12], K [13,14] and Cu [15–17], the optical and electrical properties of the obtained NiO thin films could be enhanced/improved substantially. Namely, enhanced transparency, significantly lower resistivity and sufficiently large carrier concentrations were successfully achieved. In addition to improving the optoelectronic properties of NiO-based thin films, the mechanical properties are also of critical importance when designing and fabricating the practical devices. Therefore, understanding the correlations between the mechanical properties and microstructure of NiO-based films has been of great interest. In particular, it has been widely conceived

that the wide variety of methods used for fabricating NiO thin films often resulted in very different film microstructures or even stoichiometries.

In this respect, nanoindentation has been widely used as a depth-sensing technique to measure fundamental mechanical properties such as the hardness, elastic modulus and the plasticity behaviors of various nanostructured materials [18–21] and thin films [22–26], due to its high sensitivity and excellent resolution. For instance, Fasaki et al. [27] recently investigated the effects of substrate temperature ($T_s$) on the microstructure and nanomechanical properties of NiO thin films deposited on thermally oxidized Si substrates by using pulsed laser deposition. They found that the film microstructure evolved from (200)-texturing dominant at $T_s$ = 100 °C, to (111)-texturing dominant at $T_s$ = 300 °C, and finally turned into a mixture comprising both (111)- and (200)-oriented crystallites at $T_s$ = 400 °C. Interestingly, an accompanied increase in both film hardness and elastic modulus with reduced surface roughness was also observed [27]. However, reports on how the film microstructure and nanomechanical properties are affected by metallic doping are largely lacking. On the other hand, surfaces with low free energy can reduce adhesion of the airborne contaminants, which then is effectively removed by the rolling drops due to hydrophobic behaviors [28]. Such a hydrophobic surface characteristic can potentially improve environmental durability, and hence has become one of the critical necessitated factors in many optoelectronic devices applications [29,30].

The present work aims at investigating how the nanomechanical properties and the surface wettability of the CNO thin films deposited on glass substrates by radio-frequency magnetron sputtering change with the post-deposition annealing temperatures. In particular, the effects of annealing temperature on the evolution of films microstructure and the associated surface energy modification and nanomechanical properties of the CNO thin films revealed by the wettability and nanoindentation tests are discussed.

## 2. Materials and Methods

### 2.1. Thin Films Deposition

The Cu-doped NiO (CNO) films used in the present study were deposited on glass substrates at ambient temperature by radio frequency magnetron sputtering (rf-sputtering) method. The substrates were cleaned by immersing them sequentially in ultrasonic baths of ethanol, acetone, and D.I. water for 20 min each. Prior to loading onto the substrate holder (home-made), the substrates were purged dry by pure nitrogen gas. The CNO targets (home-made) were prepared by blending copper oxide and nickel oxide powders with the molar ratio of 1:9. The mixed powder was then ball-milled with alcohol for 12 h and calcined at 900 °C for 6 h. The calcined powder was pressed into a disk of about 2″ in diameter and 2 mm in thickness to be used as the target for subsequent rf-sputtering process. However, in order to obtain more complete solid-state reaction, the pressed disks were further sintered at 900 °C for additional 24 h followed by slow cooling to room temperature at a cooling rate of 1 °C/min. The CNO films were deposited on glass substrates (Eagle XG, Corning, NY, USA) with rf-power of 100 W at the ambient temperature. Prior to deposition, the base pressure of the deposition chamber was pumped down to about $5 \times 10^{-6}$ torr. During sputtering 4 m Torr of pure Ar was charged into the sputtering chamber to serve as the working gas. The thic'kness of the obtained CNO thin films was about 200 nm. The films were then annealed at various temperatures, ranging from 300 to 500 °C, for 20 min.

### 2.2. Characterization Techniques-XRD, AFM, XPS and SEM

Crystal structures of the samples were examined by glancing incident X-ray diffraction (GIXRD) at an incident angle of 1° using a PANalytical (Malvern, UK) X'PERT PRO apparatus and Cu-K$\alpha$ radiation. The surface roughness and morphological features of CNO thin films were analyzed by atomic force microscopy (AFM; Topometrix-Accures-II, Topometrix Corporation, Santa Clara, CA, USA) and scanning electron microscopy (SEM, Hitachi S-4700, Tokyo, Japan). In addition, the chemical

state in CNO thin films was investigated by using X-ray photoelectron spectroscopy (XPS) using a PHI (Chanhassen, MN, USA) Quantum 2000 Scanning ESCA Microscopy instrument.

## 2.3. Nanoindentation

All the nanoindentation measurements were conducted at room temperature using the MTS NanoXP® system (MTS Corporation, Nano Instruments Innovation Center, Oak Ridge, TN, USA). The resolutions of the loading force and displacement are 50 nN and 0.1 nm, respectively. A Berkovich diamond indenter was pressed into CNO thin films up to a depth of 60 nm. The strain rate was varied from 0.01 to 1 s$^{-1}$. An additional harmonic movement, with amplitude and frequency being set at 2 nm and 45 Hz, respectively, was simultaneously applied on the indenter to perform the continuous stiffness measurements (CSM) [31]. Prior to conducting each test, it is important to make sure that the thermal drift is maintained to below 0.01 nm/s. In order to obtain statistical significance, at least 20 indents were conducted on each sample.

In nanoindentation experiments, the hardness is defined as $H = P_m/A_p$, with $A_p$ being the projected contact area between the indenter and the films surface and $P_m$ being the maximum indentation load at which $A_p$ is determined. For a perfectly sharp Berkovich indenter tip, the projected area is given by $A_p = 24.56 h_c^2$ with $h_c$ being the contact depth. The elastic modulus of the sample is calculated using the Sneddon [32] relation: $S = 2\beta E_r \sqrt{A_p}/\sqrt{\pi}$. Wherein, $S$ is the contact stiffness of the material and $\beta$ is a geometric constant ($\beta \approx 1$ for a Berkovich indenter tip), respectively. The reduced elastic modulus ($E_r$) is determined by the following Equation:

$$E_r = \left( \frac{1 - v_f^2}{E_f} + \frac{1 - v_i^2}{E_i} \right)^{-1} \tag{1}$$

where $v$ is the Poisson's ratio and the subscripts, $i$ and $f$, are denoted the parameters for the indenter and measured thin films, respectively. For diamond indenter tip used in the present study, $E_i$, $v_i$, and $v_f$ are taken as $E_i$ = 1141 GPa, $v_i$ = 0.07 and, $v_f$ = 0.25, respectively.

## 2.4. Wettability and Surface Energy

The contact angles were measured using a Ramehart (Succasunna, NJ, USA) Model 200 goniometer with deionized water as the liquid at room temperature. In addition, the surface wettability experiment often serves as the most convenient measure of surface energy and is commonly quantified by relating the surface energies with the obtained $\theta_{CA}$ (contact angle) via Young's equation: $\gamma_{sv} = \gamma_{sl} + \gamma_{lv} \cos\theta_{CA}$, with $\gamma_{sv}$, $\gamma_{sl}$, and $\gamma_{lv}$ being the surface tension between solid-vapor, solid-liquid, and liquid-vapor, respectively [33]. By considering that the critical interaction is the dispersive force or the van der Waals force across the interface existing between the water droplet and the solid surface, the surface energy for CNO thin films was calculated using the Fowkes-Girifalco-Good (FGG) theory [33] in combination with Young's equation. The FGG equation is given as: $\gamma_{ls} = \gamma_s + \gamma_l - 2\sqrt{\gamma_s^d \gamma_l^d}$, with $\gamma_l^d$ and $\gamma_s^d$ being denoted as the dispersive portions of the surface tension for the liquid and solid surfaces, respectively. By substituting the FGG relation into the Young's equation and employing nonpolar liquid deionized water (72.8 mJ/m$^2$) as the testing liquid, $\gamma_l^d$ is equal to $\gamma_l$, the equation becomes as following:

$$\gamma_s^d = \frac{1}{4} \gamma_l \left( \cos\theta_{CA} + 1 \right) \tag{2}$$

with $\gamma_s^d$ being the surface energy of calculated materials.

## 3. Results

Figure 1 shows the XRD results obtained for the as-deposited CNO thin films and those annealed at 300, 400 and 500 °C, respectively. The positions of the three major diffraction peaks seen in all samples

do not show noticeable shifts with increasing annealing temperature and can be unambiguously indexed as the diffraction peaks of (111), (200) and (220) planes of the rock-salt structured NiO (JCPDS 47-1049 [34]). The calculated lattice constant $a \approx 0.4199$ nm is slightly larger than the value of 0.4168 nm obtained for pure NiO by Rooksby [35], which presumably is due to the fact that the ionic size of either $Cu^{2+}$ (~73 pm) or $Cu^+$ (~77 pm) is substantially larger than that of $Ni^{2+}$ (~69 pm) [36]. In contrast, it is rather apparent from Figure 1 that the annealing temperature does have a substantial influence on the film microstructure and crystallinity. The as-deposited CNO film is more dominantly (200)-oriented with relatively broadened diffraction peaks, indicative of poorer crystallinity and/or smaller grain size. Increasing annealing temperature, nevertheless, appears to drive the microstructure of the CNO films to become more equiaxial (i.e., the intensity of the three major diffraction peaks, (111), (200), and (220), is comparable) with much improved crystalline qualities. This observation is to compare with that reported by Jang et al. [37], where the predominant growth orientation of NiO films deposited on Si(100) substrates by rf-sputtering was found to switch from (111)- to (200)-oriented as the substrate temperature ($T_s$) was raised from 200 to 300 °C. The driving force for the change has been attributed to the fact that NiO(200) surface is non-polar and has the lowest surface energy of ~1.74 J/m², compared with ~4.28 J/m² for polar (111)-terminated NiO surface [38]. On the other hand, Fasaki et al. [27] reported that NiO films deposited on thermally oxidized Si substrates by pulsed laser deposition method were (200)-oriented dominant at $T_s$ = 100 °C, which turned into (111)-oriented dominant at $T_s$ = 300 °C and finally became mixture texturing of (200)- and (111)-oriented at $T_s$ = 400 °C. Thus, it appears that texturing change due to different growth temperatures is not an unusual phenomenon. Especially, when the film is growing at elevated substrate temperatures, simultaneous annealing might also be taking place. Consequently, the eventual film orientation should be a result of complicated self-optimization among parameters such as substrate, temperature, surface energy associated with respective crystallographic orientation, grain size, strain, and etc. In our case here, since the driving force is mainly the annealing temperature, thus, the reduction of grain boundary area, relief of defect-related strain, and the surface energy associated with respective crystallographic orientation are believed to be the more relevant parameters. In this respect, Cu-doping might have introduced additional strain effects in the CNO films, which in turn may change the surface and mechanical properties manifested in the resultant films. In order to explore the possible effects resulted from Cu-doping, more quantitative analyses of the data are conducted and discussed below.

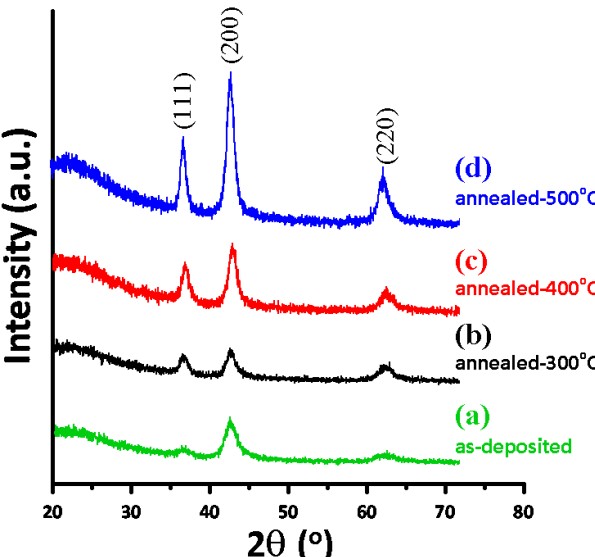

**Figure 1.** XRD patterns of CNO thin films with various annealing temperatures: (**a**) as-deposited, (**b**) 300 °C, (**c**) 400 °C and (**d**) 500 °C.

Assuming a homogenous strain across all CNO thin films, the average crystalline size ($D_S$) can be estimated from the full-width at half-maximum (FWHM) of the (200) diffraction peak by the Scherrer's equation [39], $D_S = 0.9\lambda/(\beta\cos\theta)$. Here $\lambda$ is the wavelength of the X-ray radiation (Cu K$\alpha$, $\lambda = 1.5406$ Å), $\theta$ is the Bragg angle and $\beta$ is the angle span of the FWHM of (200) peak. In the Scherrer's equation, the crystalline size is determined by assuming no lattice distortion. However, as indicated above, in the present CNO films, the lattice parameter is slightly larger than the value obtained for bulk pure NiO [34,35]. The slight peak shift due to the local strain reflecting the atomic displacements with respect to their reference positions in ideal crystal has been incorporated into the Sherrer's equation using the Williamson-Hall (W-H) analysis [40,41] to take into account both crystalline size ($D_{WH}$) and microstrain ($\varepsilon$).

The W-H equation can be depicted in the following expression [40,41]:

$$\beta\cos\theta = \frac{0.9\lambda}{D_{WH}} + 4\varepsilon\sin\theta \tag{3}$$

In Figure 2, the ($\beta\cos\theta$)/$\lambda$ versus ($\sin\theta$)/$\lambda$ plots based on Equation (3) are presented. The y-intercept and the slope of the each curve give the values of reciprocal $D_{WH}$ and $\varepsilon$ of each film, respectively. The obtained values of grain sizes using Scherrer's original equation and W-H equation (Equation (3)) are denoted as $D_S$ and $D_{WH}$, respectively, and are listed in Table 1 together with the obtained microstrain.

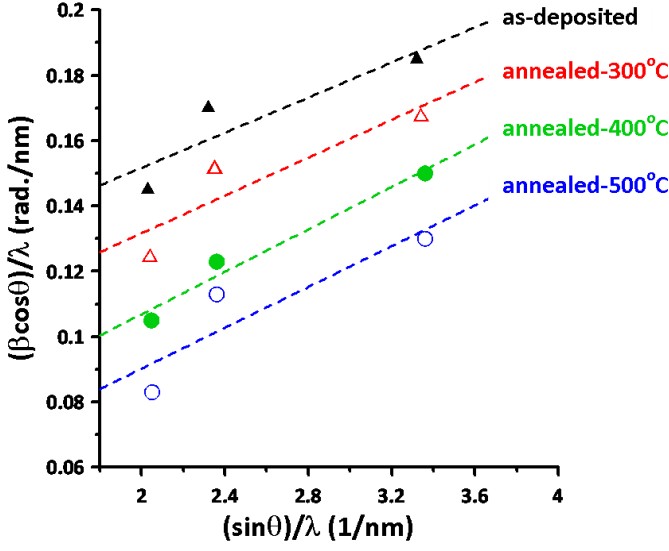

**Figure 2.** Williamson-Hall plots for measured XRD data of as-deposited and annealed CNO thin films.

**Table 1.** The microstructural parameters of CNO thin films evaluated from XRD results.

| Sample | $D_S$ (nm) | $D_{WH}$ (nm) | $\varepsilon$ (%) |
|---|---|---|---|
| as-deposited CNO film | 5.7 | 10.3 | 0.65 |
| annealed-300 °C CNO film | 8.4 | 13.6 | 0.70 |
| annealed-400 °C CNO film | 11.2 | 23.8 | 0.77 |
| annealed-500 °C CNO film | 18.6 | 38.5 | 0.80 |

It is noted from Table 1 that the incorporation of microstrain effect has evidently resulted in significant differences in the values of grain size. The grain size calculated using the original Scherrer's equation systematically gives values smaller than those calculated using Equation (3), suggesting that the microstrain is indeed significantly affecting the XRD results. Moreover, the results also showed that both the values of $D_S$ and $D_{WH}$ increase with increased annealing temperature, indicating that

the overall crystallinity of CNO thin films were substantially improved, presumably due to higher thermal energy provided.

The morphological features of all CNO thin films are observed by SEM, as shown in Figure 3. It is evident that all films have uniformly distributed spherical granular microstructures. The feature of granular surface morphology, nevertheless, is enhanced significantly when annealing temperature is increased from 300 to 500 °C, manifested by the increased diameter of the spherical grains. The average grain size is estimated to range from 10 nm to 40 nm, which is in good agreement with the XRD analysis.

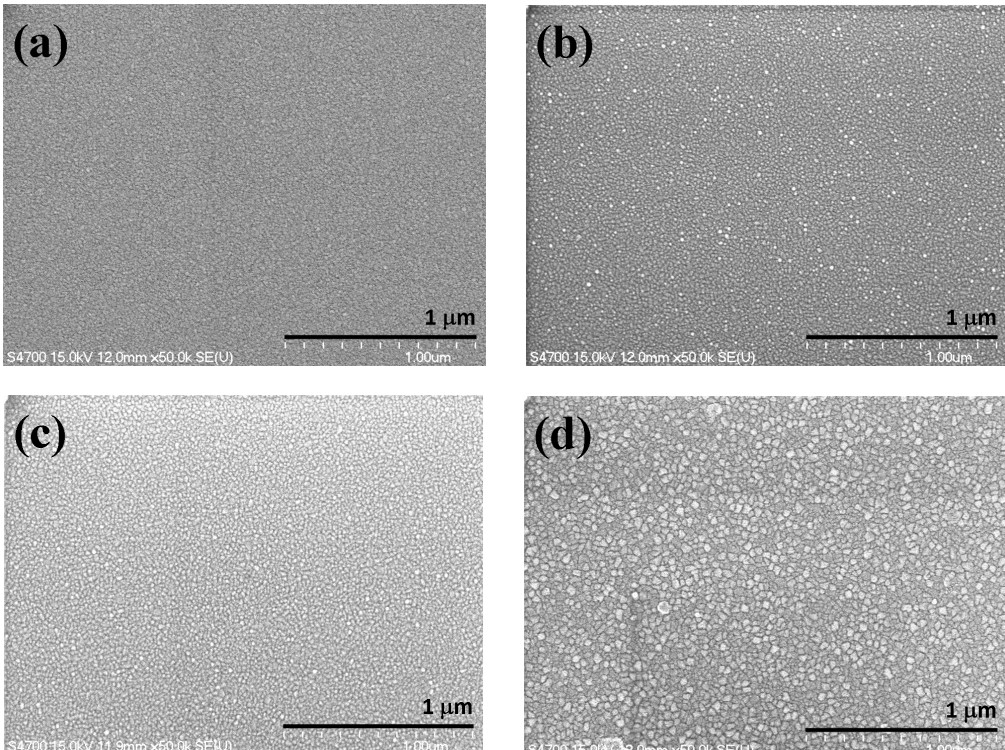

**Figure 3.** SEM images of: (**a**) as-deposited CNO thin film and, annealed at (**b**) 300 °C, (**c**) 400 °C and (**d**) 500 °C CNO thin films.

To further elucidate whether Cu-doping is homogeneous, we also performed XPS measurements to examine the valence state of Ni and Cu elements. Figure 4 shows the XPS spectra of Ni $2p_{3/2}$, O $1s$ and Cu $2p_{3/2}$ of as-deposited and annealed CNO thin films, respectively. Although at the first glance of the obtained Ni $2p_{3/2}$ spectra, it appears that the general features of the spectra are not influenced noticeably by annealing; more detailed analyses and comparisons with previous studies are, nevertheless, necessary. According to Kitakatsu et al. [42], the first peak centering around $852 \pm 0.1$ eV with its satellite at $858.5 \pm 0.1$ eV is ascribed to metallic Ni; a doublet at $854.1 \pm 0.1$ and $855.9 \pm 0.1$ eV with a satellite at $861.1 \pm 0.1$ eV are assigned to $Ni^{2+}$ state in NiO; a third small, but necessary contribution at $856.0 \pm 0.1$ eV with a satellite at $862.1 \pm 0.1$ eV is arising from the hydroxylated $Ni^{2+}$. Several subsequent studies [43–46] had similarly assigned the peak around 853.8–854.5 eV to $Ni^{2+}$ state in the standard Ni-O octahedral bonding configuration in the cubic rocksalt NiO structures. Another peak around at 855.7–857.4 eV was ascribed to $Ni^{3+}$ ($Ni_2O_3$) or nickel hydroxides. In addition, the satellite peaks of the binding energies of NiO and $Ni_2O_3$ [45]) were also found to locate around 860–862 eV, which were in very good agreement with the assignments given by Kitakatsu et al. [42].

For the O $1s$ XPS spectra in NiO, the signals could in general be decomposed into two (low and high energy) components: namely, $529.6 \pm 0.1$ and $531.4 \pm 0.1$ eV for $O^{2-}$ in NiO and hydroxyl (OH–) groups, respectively. Additional peak centering around $533.2 \pm 0.4$ eV is often assigned to the existence of $H_2O$ at the sample surface [41]. By comparing the XPS spectra of Ni $2p_{3/2}$ (left column of Figure 4)

and O 1*s* (middle column of Figure 4) obtained from the present study with those described above, it is evident that the hydroxylation of Ni is much more pronounced when the annealing temperature was raised to 400 °C and above.

On the other hand, as pointed out by Martin et al. [44], it is much more difficult to differentiate the valence state of Cu and $Cu^+$ by simply resorting to the core position of Cu $2p_{3/2}$ owing to the very close peak positions located at 932.6 and 932.8 eV for Cu and $Cu^+$, respectively. Nevertheless, for $Cu_2O$ ($Cu^+$), a very weak satellite peak at higher binding energies, between 942.0 and 948.0 eV, can be detected, which is absent for metallic Cu. For CuO samples, the main $2p_{3/2}$ peak consists of two components, the main one located at 933.6 eV and the low intensity one located at 932.8 eV (corresponds to $Cu^+$) [44].

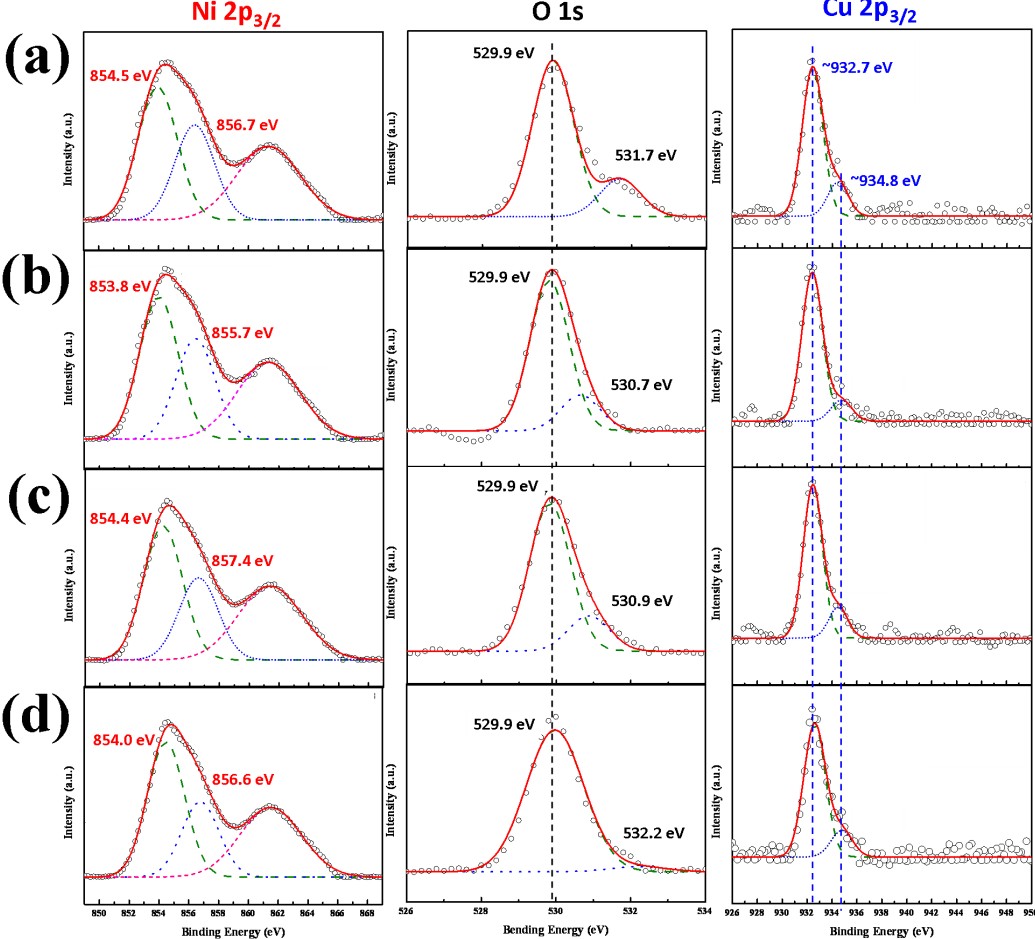

**Figure 4.** XPS spectra of Ni $2p_{3/2}$ (left column), O 1s (middle column) and Cu $2p_{3/2}$ (right column) of as-deposited and annealed CNO thin films, respectively. Conditions: (**a**) as-deposited, (**b**) annealed-300 °C, (**c**) annealed-400 °C and (**d**) annealed-500 °C.

As shown in the right column of Figure 4, the Cu $2p_{3/2}$ XPS for the present samples, nevertheless, are relatively noisy, presumably due to the much smaller concentration as compared to Ni and oxygen. Thus, it is not very conclusive as to whether or not there exist satellite peaks at higher binding energies (between 942.0 and 948.0 eV). The main peak, however, can be de-convolved into two peaks at 932.7 and 934.8 eV, respectively, which can be approximately assigned to the binding energies of $Cu^+$ and $Cu^{2+}$ peaks. It is noted that the peaks remained essentially unaffected by the annealing temperature. Moreover, from the relative intensities of the de-convolved peaks, it is suggestive that the ratio of $Cu^+/(Cu^+ + Cu^{2+})$ is over 80%, indicating that the incorporated Cu in the present thin films is largely existing as $Cu^+$. The observed ratio of $Cu^+/(Cu^+ + Cu^{2+})$ is in contrast to that reported by Sato et al. [45],

wherein $Cu^+/(Cu^+ + Cu^{2+}) \approx 35\%$ was obtained in dc-sputtered NiO films with similar Cu doping (~9.6%). These results, however, are consistent with the expanded lattice constant and significant strain effects obtained from the XRD analyses discussed above.

To further explore how the structural change originated from Cu-doping and thermal annealing affects the film surface morphology and associated properties, the surface features of CNO thin films are examined by AFM; the results are displayed in Figure 4. It is evident from the AFM images that all CNO films exhibit homogenous microstructures, albeit the root-mean-square surface roughness ($R_{rms}$) appears to increase progressively with increasing annealing temperatures. The $R_{rms}$ values are 0.7, 1.4, 2.9 and 3.8 nm for the as-deposited CNO film and those annealed at 300, 400 and 500 °C, respectively. This observation is also in line with the increased grain size and, perhaps, texturing switching phenomena induced by increasing the annealing temperature discussed above. It is interesting to note that, as displayed in the insets of Figure 5a–d, the wettability behavior of the surface is also strongly affected by the morphologies of the sample surface. As a result, it is expected that water droplets on the surface with rougher $R_{rms}$ should have larger $\theta_{CA}$ [46]. Indeed, as displayed in the insets of Figure 5a–d, the wettability experiments evidently give the $\theta_{CA}$ values of 45.7°, 55.8°, 80.4°, and 97.5° for the as-deposited CNO film and those annealed at 300, 400 and 500 °C, respectively. The corresponding $R_{rms}$ are 0.7, 1.4, 2.9 and 3.8 nm, respectively. Alternatively, as mentioned above, since the film texturing orientation was also found to change with the annealing temperature, the corresponding surface energy might play a role, as well.

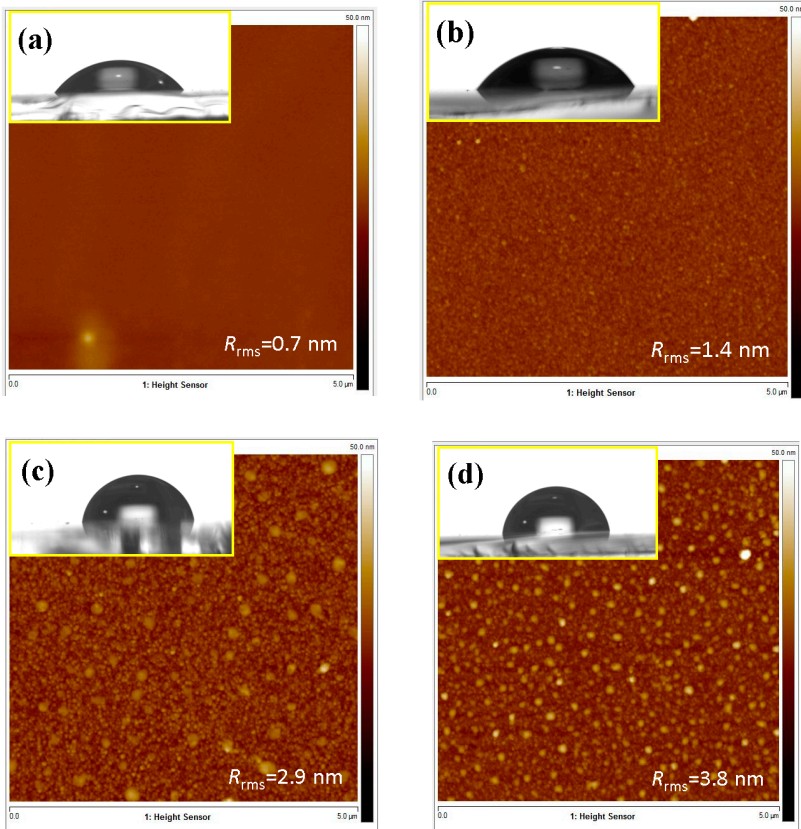

**Figure 5.** AFM images of: (**a**) as-deposited, (**b**) 300 °C, (**c**) 400 °C and (**d**) 500 °C, CNO thin films. Insert: the corresponding contact angle of 45.7°, 55.8°, 80.4° and 97.5°, respectively.

Further, by straightforward analyzes, the values of surface energy obtained for as-deposited CNO and those annealed at 300, 400 and 500 °C are 30.9, 28.4, 21.2 and 15.8 mJ/m², respectively. We note that these values are much smaller than the intrinsic surface energy of the non-polar NiO(200) surface (~1.74 J/m²) [38], presumably due to the substantial modification of the surface condition, such as

the hydroxylation revealed in the XPS data discussed above. As a consequence, it is conceived that the annealing-driven changes in crystallographic orientation texturing might not be a relevant factor for the observed surface property changes. Nevertheless, the annealing-induced trend of driving the film to become more hydrophobic is clearly evident. Extrinsically, in addition to the effect of surface hierarchical modification (namely the increased surface roughness), described above [47], the improved crystallinity, and hence reduction of the defect concentration, may also play a role, as well. Indeed, as pointed out by Bayati et al. [48], point defects, such as oxygen vacancies, were mainly responsible for the significant hydrophilicity (contact angle ~22°) observed in their laser pulses irradiated yttria stablilized zirconia films. In our case here, the removal of the grown-in defects during deposition by annealing, thus, is expected to result in increasing surface hydrophobicity.

Next, we turn to discuss the effects of annealing temperature on the mechanical properties of the CNO films revealed by nanoindentation measurements. The typical CSM load-displacement curves of the CNO thin films annealed at the various temperatures are shown in Figure 6.

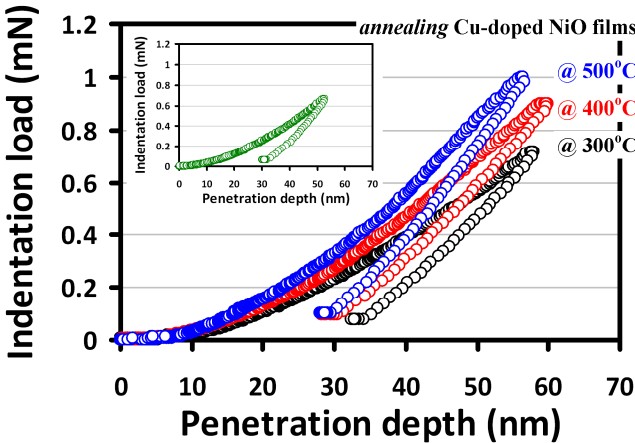

**Figure 6.** The load-displacement curves of CNO thin films with various annealing temperatures. Insert: the load-displacement curve of as-deposited CNO film.

The inset of Figure 6 shows the load-displacement curve of the as-deposited CNO film for comparison. In general, the nanoindentation curves can provide rich information about the elastic and plastic deformation behaviors of the materials, and prominent parameters such as hardness and the Young's modulus can be readily obtained [49,50]. Moreover, it is suggested that, when performing the nanoindentation tests, the nanoindentation depth should never exceed 30% of the films' thickness or the size of nanostructures [51]. Here, the total penetration depth into the CNO thin film was approximately 60 nm, compared to the film thickness of ~200 nm, which is well within the abovementioned criterion for reliable nanoindentation tests. Following the analytical procedures described previously [22,50], the values of hardness (and Young's modulus) obtained from the load-displacement curves of the as-deposited and the post-annealed CNO thin films displayed in Figure 6 are $16.2 \pm 0.4$ ($154.8 \pm 10.4$), $15.4 \pm 0.2$ ($149.6 \pm 11.2$), $21.5 \pm 0.3$ ($165.8 \pm 12.5$) and $25.2 \pm 0.6$ ($206.5 \pm 11.9$) GPa, respectively. The above results are plotted as a function of annealing temperature in Figure 7. A clear increasing tendency of both hardness and elastic modulus with increasing annealing temperature is evident, except for the as-deposited film. Compared to the obtained changing tendency of grain size listed in Table 1, the results appear to follow the notion of the inverse Hall-Petch effect [52]. In contrast to the conventional Hall-Petch effect, where the activities of dislocations play the primary role, in the inverse Hall-Petch effect regime, the grain boundary sliding is the prominent mechanism in determining film hardness and mechanical strength [53]. It is not surprising if one considers that the grain sizes of the present CNO films are all in the order of few tens of nanometers, while for most metals, where the conventional Hall-Petch relations are ubiquitously applicable, the grain sizes are all well above order of micrometers. Finally, it is noted that the film's internal strain also shows a similar

trend with increasing annealing temperature (see Table 1), and may also partially contribute to the observed increased film hardness and elastic modulus. Annealing could have thermally assisted the doped Cu ions to occupy the Ni sites, and the ionic size difference, in turn, gives rise to the internal lattice strain. Within the context of this scenario, the slight decrease of the hardness between the as-deposited and 300 °C-annealed CNO thin films may be explained by the relaxation of residual stress via recrystallization and crystallite reorientation during the early stage of the annealing process [49].

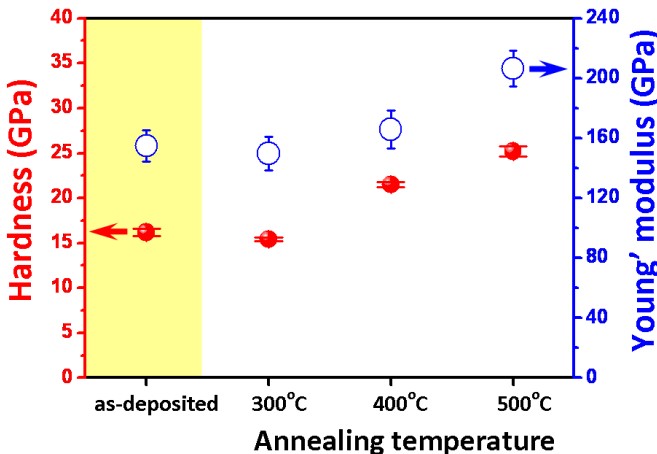

**Figure 7.** Hardness and Young's modulus of CNO thin films with various conditions.

## 4. Conclusions

In summary, we conducted a comprehensive investigation on the effects of post-annealing on the microstructure, surface, and nanomechanical properties of CNO thin films deposited on glass substrates by the rf-sputtering method. The main findings are briefly summarized as following:

- The XRD, AFM and SEM results consistently indicated that the crystalline size ($D_S$ and $D_{WH}$), surface roughness, and internal strain of the CNO films are significantly increased with increasing annealing temperature.
- The Ni $2p_{3/2}$ and O $1s$ XPS spectra indicated that the films might have substantial hydroxylation during the annealing processes. On the other hand, Cu $2p_{3/2}$ suggested that the chemical bonding state of Cu ions in CNO thin films is largely of Cu$^+$ characteristics, resulting in significant internal strain due to a larger ionic size difference between Cu$^+$ (77 pm) and Ni$^{2+}$ (69 pm).
- The contact angle ($\theta_{CA}$) is increased from 45.7° for the as-deposited films to 97.5° for films annealed at 500 °C. In addition, the obtained surface energies at various states (~few tens of mJ/m$^2$), however, are order of magnitude smaller than that of the intrinsic (200)-NiO surface (~1.74 J/m$^2$), indicating that the surface properties are more relevant to the extrinsic morphological factors.
- The hardness (Young's modulus) of CNO thin films are increased from 15.4 ± 0.2 (149.6 ± 11.2) GPa to 25.2 ± 0.6 (206.5 ± 11.9) GPa by increasing the annealing temperature from 300 °C to 500 °C from the nanoindentation results, indicating that the primary deformation mechanism in these films is grain boundary sliding instead of dominated by the dislocation-glidings.

**Author Contributions:** Conceptualization, S.-H.W. and S.-R.J.; Methodology, S.-H.W., G.-J.C. and H.-Z.C.; Formal Analysis, S.-H.W., G.-J.C. and H.-Z.C.; Investigation, S.-H.W.; Writing—Original Draft Preparation, S.-R.J.; Writing—Review & Editing, S.-R.J. and J.-Y.J.

**Funding:** The authors are thankful for the financial support of the Ministry of Science and Technology, Taiwan under Contract Nos. MOST 106-2112-M009-013-MY3, MOST 107-2112-M-214-001, MOST 106-2112-M-214-001, MOST 105-2112-M-214-001, MOST 106-2221-E-214-014 and MOST 104-2221-E-214-003.

**Acknowledgments:** The authors like to thank W.-J. Hsieh and C.-F. Yang for their technical support.

**Conflicts of Interest:** The authors declare no conflict of interest.

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
