# Peer review of "Annealing-Driven Microstructural Evolution and Its Effects on the Surface and Nanomechanical Properties of Cu-Doped NiO Thin Films"

_coatings, doi:10.3390/coatings9020107_

Round 1
Reviewer 1 Report
please find my comments below. I will try to follow the same order as the manuscript is designed.
1. Out of curiosity I would like to ask you how you cleaned the substrates prior to deposition and if you have ever noticed any effect of the cleaning on the following deposition.
2. Results.
- Could you show the graphical results of the W-H analysis?
- Could you also add any SEM or TEM analysis to confirm what you are assuming about the crystallinity of these films?
- I reckon that in the XPS analysis section you have mentioned quite a lot of literature review, that in the end is not useful to understand what you are presenting. I would focus more on the XPS analysis itself. I do not necessary agree with you on the 'evident hydroxylation of Ni' (line 201...). I would check the shape of the multiple-split Ni2p3/2 and the satellite in the range 870-880 eV , they are actually very distinctive for NiO and Ni(OH)2 (see fig. below). Please provide these data as well, if you can.
As for O1s spectrum , I would double check it and make sure it is not NiOx
Same consideration for Cu2p3/2, consider a larger eV range (see the attachment from http://www.xpsfitting.com/2012/01/cu0cuii-or-cuicuii-calculations.html) to be completely sure about the valence state of Cu.
- Can you also introduce undoped NiO coatings as a reference, to understand the effect of the the Cu-doping? and the further question would be, have you tried to study and compare different Cu-doping concentrations?
-lines 237-240, I do not understand how relevant these materials (TiO2 and ZnO) are to your study. As mentioned before, I would avoid to much literature review in the discussion, I would support it with more results instead.
- I would move the theoretical part of the FGG theory in combination with Young's equation, in the Materials and Methods section.
- I would also try to discuss mainly your experimental data and find conclusions out of what you have actually done. There are assumptions that are not proved by your experimental work. Either you implement it with more analyses or rewrite the conclusions and discussion strictly following the experimental data that you have got.
I believe that the work is valuable. The manuscript needs to be revisited a bit.
Good luck with your work and look forward to reading it soon.
Kind regards

Author Response
Please kindly find the attached file. "Rebuttal_coatings-429525_1"
Reviewer #1
1. Out of curiosity I would like to ask you how you cleaned the substrates prior to deposition and if you have ever noticed any effect of the cleaning on the following deposition.
Responses: The substrates were cleaned by immersing them sequentially in ultrasonic baths of ethanol, acetone, and, D.I. water for 20 minutes each. Prior to loading onto substrate holder, the substrates were purged dry by pure nitrogen gas. We did not notice any substantial effects from the cleaning processes on the subsequent deposition.
2. Results.
- Could you show the graphical results of the W-H analysis?
Responses: We have followed the reviewers’ suggestion to add the new W-H plot and the relevant sentences in the revised manuscript.
- Could you also add any SEM or TEM analysis to confirm what you are assuming about the crystallinity of these films?
Responses: We have followed the reviewers’ suggestion to add the SEM images of all CNO thin films and relevant sentences in the revised manuscript.
- I reckon that in the XPS analysis section you have mentioned quite a lot of literature review, that in the end is not useful to understand what you are presenting. I would focus more on the XPS analysis itself. I do not necessary agree with you on the 'evident hydroxylation of Ni' (line 201...). I would check the shape of the multiple-split Ni2p3/2 and the satellite in the range 870-880 eV, they are actually very distinctive for NiO and Ni(OH)2 (see fig. below). Please provide these data as well, if you can.
Responses: For Ni2p spectrum, Kitakatsu et al. [#1] reported that a doublet at 854.1 ±0.1 and 855.9 ±0.1 eV with a satellite at 861.1 ±0.1 eV assigned to the Ni 2+ state in NiO. Meanwhile, the high binding energy shoulder of the O ls signal and the small contribution at 856.7+0.1 eV that could be introduced in the Ni 2p3/2 signal indicate the presence of OH- groups bonded to Ni 2+ cations. Obviously, our experimental data is consistent with those reported in Ref [#1]. Therefore, we concluded that the existence of the hydroxylated Ni 2+.
[#1] N. Kitakatsu, V. Maurice, C. Hinnen, P. Marcus, Surf. Sci. 407 (1998) 36.
As for O1s spectrum , I would double check it and make sure it is not NiOx. Same consideration for Cu2p3/2, consider a larger eV range (see the attachment from http://www.xpsfitting.com/2012/01/cu0cuii-or-cuicuii-calculations.html) to be completely sure about the valence state of Cu.
Responses: If the films are NiOx, there might be a doublet Ni2p3/2 peak at about 853.7 and 857 eV, yet it is quite different from our Ni2p data. Therefore, we excluded the possibility of NiOx in the CNO films. As for Cu2p3/2 spectrum, both the ratio of Cu1+/(Cu1++Cu2+) and the corresponding lattice expansion explain the existence of Cu1+ and Cu2+.
- Can you also introduce undoped NiO coatings as a reference, to understand the effect of the Cu-doping? and the further question would be, have you tried to study and compare different Cu-doping concentrations?
Responses: In this manuscript, the annealing parameters on the microstructure and nanomechanical properties of the CNO films were concerned. The effects of various Cu doping levels on the CNO films would be our ongoing works.
-lines 237-240, I do not understand how relevant these materials (TiO2 and ZnO) are to your study. As mentioned before, I would avoid too much literature review in the discussion, I would support it with more results instead.
Responses: We are very grateful to Reviewer’s comment. In order to avoid confusing the reader, the sentences are removed from the revised manuscript.
- I would move the theoretical part of the FGG theory in combination with Young's equation, in the Materials and Methods section.
Responses: We have followed the reviewers’ suggestion to move the FGG theory to “Materials and Methods section”.
- I would also try to discuss mainly your experimental data and find conclusions out of what you have actually done. There are assumptions that are not proved by your experimental work. Either you implement it with more analyses or rewrite the conclusions and discussion strictly following the experimental data that you have got.
Responses: The authors have rewritten the conclusion and discussions according reviewers’ comments.
Reviewer 2 Report
The paper deals with characterisation of NiO doped by Cu thin films prepared by rf sputtering from oxide target. The main concern of this work is to study impact of post annealing temperature on structural, morphological, and nanomechanical properties of deposited thin films. The manuscript is written in proper manner and in expected logical structure. I think that the paper has not critical weaknesses and could be published as is.
Author Response
Please kindly find the attached file. "Rebuttal_coatings-429525_2"
Reviewer #2
The paper deals with characterisation of NiO doped by Cu thin films prepared by rf sputtering from oxide target. The main concern of this work is to study impact of post annealing temperature on structural, morphological, and nanomechanical properties of deposited thin films. The manuscript is written in proper manner and in expected logical structure. I think that the paper has not critical weaknesses and could be published as is.
Responses: We are very grateful for receiving the positive and encouraging comments from Reviewer #2.
Reviewer 3 Report
Comments:
The authors describe annealing effect of Cu-doped NiO on microstructural and nanomechanical properties. The reviewer has some comments to improve the quality of manuscript for publication in Coatings Journal.
1. For Fig. 2, this part is too long and hard to get points (the points are scattered). It should be focused on the authors’ results first and shorten. The authors need to describe deconvolution peaks for each graph and then how those peaks change with annealing temperature and why?
2. Pls, add the AFM roughness value on Fig. 3.
3. Pls, add the symbols on Fig. 5.
Author Response
Please kindly find the attached file. "Rebuttal_coatings-429525_3"
Reviewer #3
The authors describe annealing effect of Cu-doped NiO on microstructural and nanomechanical properties. The reviewer has some comments to improve the quality of manuscript for publication in Coatings Journal.
1. For Fig. 2, this part is too long and hard to get points (the points are scattered). It should be focused on the authors’ results first and shorten. The authors need to describe deconvolution peaks for each graph and then how those peaks change with annealing temperature and why?
Responses: We have rewritten the paragraphs discussing XPS results (Fig. 4 in the revised manuscript) to comply with reviewers’ comments.
2. Pls, add the AFM roughness value on Fig. 3.
Responses: We have followed the reviewers’ suggestion to modify the AFM figures.
3. Pls, add the symbols on Fig. 5.
Responses: We have followed the reviewers’ suggestion to modify the figure.
Round 2
Reviewer 3 Report
Manuscript ID: coatings-429525R
Title: Annealing-driven microstructural evolution and its effects on the surface and nanomechanical properties of Cu-doped NiO thin films
Comments:
The authors describe the annealing effect of Cu-doped NiO on microstructural and nanomechanical properties. The manuscript is improved and it can be published as is.
Author Response
Thank you very much for you effort.